# PrEP Scale-Up and PEP in Central and Eastern Europe: Changes in Time and the Challenges We Face with No Expected HIV Vaccine in the near Future [note 1]

**DOI:** 10.3390/vaccines11010122

**Published:** 2023-01-04

**Authors:** Deniz Gokengin, Dominik Bursa, Agata Skrzat-Klapaczynska, Ivailo Alexiev, Elena Arsikj, Tatevik Balayan, Josip Begovac, Alma Cicic, Gordana Dragovic, Arjan Harxhi, Kerstin Aimla, Botond Lakatos, Raimonda Matulionyte, Velida Mulabdic, Cristiana Oprea, Antonios Papadopoulos, Nino Rukhadze, Dalibor Sedlacek, Lubomir Sojak, Janez Tomazic, Anna Vassilenko, Marta Vasylyev, Antonija Verhaz, Nina Yancheva, Oleg Yurin, Justyna Kowalska

**Affiliations:** 1Department of Infectious Diseases and Clinical Microbiology, Medical Faculty, Ege University, Izmir 35100, Türkiye; 2HIV/AIDS Research and Practice Center, Ege University, Izmir 35100, Türkiye; 3Department of Adults’ Infectious Diseases, Hospital for Infectious Diseases, Medical University of Warsaw, 01-201 Warsaw, Poland; 4Department of Virology, National Center of Infectious and Parasitic Diseases, 1504 Sofia, Bulgaria; 5University Clinic for Infectious Diseases and Febrile Conditions Skopje, Faculty of Medicine Skopje, Ss.Cyril and Methodius University, 1010 Skopje, North Macedonia; 6National Center for Disease Control and Prevention, Yerevan 0025, Armenia; 7Department of Infectious Diseases, School of Medicine, University of Zagreb, 10000 Zagreb, Croatia; 8Center for Communicable Disease Control and Prevention, Institute for Public Health of Montenegro, 81000 Podgorica, Montenegro; 9Department of Pharmacology, Clinical Pharmacology and Toxicology, School of Medicine, University of Belgrade, 11000 Belgrade, Serbia; 10Department of Infectious Disease, Faculty of Medicine, University Hospital Center of Tirana, 1001 Tirana, Albania; 11Department of Infectious Diseases, Tartu University Hospital, 50406 Tartu, Estonia; 12Department of HIV and Tropical Diseases, South Pest Central Hospital, National Institute of Hematology and Infectious Diseases, 1097 Budapest, Hungary; 13Department of Infectious Diseases and Dermatovenerology, Vilnius University Hospital Santaros Klinikos, Vilnius University, LT-08410 Vilnius, Lithuania; 14Clinic for Infectious Diseases, Clinical Center University of Sarajevo, 71000 Sarajevo, Bosnia and Herzegovina; 15Victor Babes Hospital for Infectious and Tropical Diseases, Carol Davila University of Medicine and Pharmacy, 030303 Bucharest, Romania; 164th Department of Internal Medicine, Medical School, National and Kapodistrian University of Athens, University General Hospital “ATTIKON”, 12462 Athens, Greece; 17Infectious Diseases, AIDS & Clinical Immunology Research Center, 0160 Tbilisi, Georgia; 18HIV Center University Hospital, Charles University, 11000 Pilsen, Czech Republic; 19Center for Treatment of HIV/AIDS Patients, Department of Infectology and Geographical Medicine, Academic L. Derer’s University Hospital, 2412 Bratislava, Slovakia; 20Department of Infectious Diseases, University Medical Center Ljubljana, 1525 Ljubljana, Slovenia; 21Global Fund Grant Management Department, Republican Scientific and Practical Center of Medical Technologies, Informatization, Management and Economics of Public Health (RNPT MT), 220013 Minsk, Belarus; 22Astar Medical Center, 79054 Lviv, Ukraine; 23Internal Medicine Department, Erasmus MC, 2040 3000 Rotterdam, The Netherlands; 24Clinic for Infectious Diseases, University Clinical Center of the Republic of Srpska, 78000 Banja Luka, Bosnia and Herzegovina; 25Department for AIDS, Specialized Hospital for Active Treatment of Infectious and Parasitic Diseases—Sofia, Medical University Sofia, 1431 Sofia, Bulgaria; 26Department of AIDS, Epidemiology and Prevention, Central Research Institute of Epidemiology, Federal AIDS Centre, 111123 Moscow, Russia; 27HIV Out-Patient Clinic, Hospital for Infectious Diseases, Medical University of Warsaw, 01-201 Warsaw, Poland

**Keywords:** HIV, pre-exposure prophylaxis, post-exposure prophylaxis, Central and Eastern Europe

## Abstract

With no expected vaccine for HIV in the near future, we aimed to define the current situation and challenges for pre- and post-exposure prophylaxis (PrEP and PEP) in Central and Eastern Europe (CEE). The Euroguidelines CEE Network Group members were invited to respond to a 27-item survey including questions on PrEP (response rate 91.6%). PrEP was licensed in 68.2%; 95 centers offered PrEP and the estimated number on PrEP was around 9000. It was available in daily (40.1%), on-demand (13.3%), or both forms (33.3%). The access rate was <1–80%. Three major barriers for access were lack of knowledge/awareness among people who are in need (59.1%), not being reimbursed (50.0%), and low perception of HIV risk (45.5%). Non-occupational PEP was available in 86.4% and was recommended in the guidelines in 54.5%. It was fully reimbursed in 36.4%, only for accidental exposures in 40.9%, and was not reimbursed in 22.72%. Occupational PEP was available in 95.5% and was reimbursed fully. Although PrEP scale-up in the region has gained momentum, a huge gap exists between those who are in need of and those who can access PrEP. Prompt action is required to address the urgent need for PrEP scale-up in the CEE region.

## 1. Introduction

Despite a decreasing trend in global HIV incidence during the last decade, Central and Eastern Europe (CEE) have shown a significant rising trend during this period [1,2]. However, the numbers seem to have been declining from 2020 to 2021, which is thought to be due to reduced testing and delays in reporting during the COVID-19 pandemic and is expected to regain the previous pace [3]. The pandemic heavily disrupted the HIV cascade of care, including screening globally and in the CEE region, which resulted in a decline in HIV diagnoses and impacted HIV care services severely [4,5]. The 90–90–90 goals set by UNAIDS in 2014 could not be met in 2020, leading to a more ambitious target of 95–95–95 being set for 2025 [6]. However, the pandemic was shown to reverse most of the gains made to date, and there is a lot more to be done in terms of HIV prevention. 

Ongoing studies for developing an HIV vaccine do not suggest a working vaccine in the near future [7]. Currently, HIV prevention focuses on a combination prevention approaches, including treatment as prevention (TaSP), undetectable = untransmittable (U = U), condom use, risk reduction strategies, and pre- and post-exposure prophylaxis (PrEP and PEP) [8]. PrEP has been proven to be highly effective at reducing the number of new infections when used in line with recommendations [9,10,11,12,13] and PEP is a life-saving strategy for incidental unprotected contact with the virus both in healthcare and non-occupational settings [14,15]. PrEP roll-out has been slow but steady in Europe after its introduction in France in 2016, with 102,446 PrEP initiations rising up to 2,797,304 in the second quarter of 2022 [16]. However, there are great inequalities in availability and access to PrEP across countries in Europe. While the number of countries where PrEP is a licensed tool for prevention has been steadily increasing, the coverage of people who are in highest need is still very low. While the overall estimated PrEP gap for the European Union (EU) countries in 2019 was 17.4%, the largest gap was calculated for non-European Union (EU) and/or CEE countries [17].

Central and Eastern Europe is a long-neglected region in terms of addressing the HIV epidemic due to the long-term low prevalence of HIV infection, and economic, cultural, social, and political diversity of the region. Earlier long-term conflicts in the region such as the breakup of former Yugoslavia into smaller states and the collapse of the former Soviet Union led to severe political and economic crises and dramatic socioeconomic changes that challenged health systems [18,19,20,21]. This resulted in an increase in injecting drug use and sex work, creating the perfect high-risk environment for HIV spread. An almost 60% increase in annual HIV infections were reported in the new states of the former Soviet Union between 2010 and 2015 [21,22,23]. Adding the impact of limited resources in several new states allocated to the response to the rapidly growing epidemic [18,21], lack of international funding and support [20], stigma against LGBTI communities [20], and punitive laws for men who have sex with men and for injecting drug use [20] led to a dramatic increasing trend in new HIV diagnoses. 

The Euroguidelines in Central and Eastern Europe (ECEE) Network Group was established in 2016 to review and improve the implementation of European guidelines in CEE and to gather this diverse group of countries under one umbrella to enhance collaboration and joint work. Its Working Group consists of 47 experts in the field of HIV and infectious diseases from 24 countries [24]. Countries included in the network are Albania, Armenia, Belarus, Bosnia and Herzegovina, Bulgaria, Croatia, Czech Republic, Estonia, Finland, Georgia, Greece, Hungary, Lithuania, Montenegro, North Macedonia, Poland, Republic of Moldova, Romania, Russian Federation, Serbia, Slovakia, Slovenia, Turkey, and Ukraine. The group published several articles since its establishment in order to report on various aspects of the HIV epidemic in regions where data are very limited.

The aim of this study was to (1) review the current status for two HIV prevention tools, PrEP and PEP, in the CEE region and to compare it with previous data from the ECEE Network group; (2) to identify potential barriers in their implementation; and (3) to discuss possible ways forward.

## 2. Materials and Methods

A 27-item survey questioning PrEP and PEP was prepared by the authors based on a previous survey run in the group in previous years [25,26]. The survey included questions on PrEP availability and accessibility, PrEP guidelines, cost of the drugs, barriers for PrEP implementation, numbers on PrEP and PEP availability, guidelines, and coverage. The survey is provided as a Appendix A. All members of the ECEE Network Group from 24 countries were invited to respond to the survey between 14 November 2021 and 24 January 2022. More than one response was received for two countries; discrepant answers were clarified by communicating directly with the respondents. 

Data were analyzed using the Statistical Package for Social Sciences 22 program. Categorical variables were given as frequency and percentages, and for continuous variables the median and interquartile range (IQR) values were calculated. The data from the 2018 survey and the current survey including only countries that responded to both surveys were compared, where only the distribution of numbers and percentages were defined; it was not possible to make a statistical comparison because the respondent numbers were very small. 

## 3. Results

Twenty-four respondents from twenty-two countries responded to the survey (91.6% response rate); the Republic of Moldova and Finland did not participate in the survey. 

PrEP was licensed in 15/22 (68.2%) countries and recommended in the national HIV guidelines in 12/22 (54.5%). Five countries had national guidelines, but they did not include any recommendation for PrEP, two countries did not have national guidelines and one country had an outdated guideline and all three used the European AIDS Clinical Society (EACS) guidelines, one country had specific PrEP guidelines, and one country stated that they did not need guidelines to prescribe PrEP. Six countries (27.3%) reported they also used other PrEP guidelines including EACS, Department of Human and Health Sciences (DHHS), and European Centers for Disease Control (ECDC) guidelines (Table 1).

The available PrEP types in 15 countries where PrEP was licensed are shown in Figure 1 and settings where PrEP was prescribed are listed in Table 1.

PrEP was available to buy in pharmacies only as a generic formulation in 6 out of 22 countries (27.2%), both as an original and generic formulation in 4 (18.2%) countries, was not available in 10 (45.5%) countries, and was available in other settings in 2 (9.1%) countries (Table 1). Turkey reported that original and generic formulations for antiretroviral treatment were available for purchase in pharmacies, but only on-demand formulations for PrEP were available as generics. The Czech Republic reported availability only from the hospital pharmacy in generic form for outpatients, and Bosnia and Herzegovina reported availability in specific pharmacies upon referral of the patient by the clinician. The median (IQR) price of the original and generic formulations was 199 (95–249) EUR and 50 (17–70) EUR, respectively.

The rate of new HIV infections diagnosed in the previous year was highest in the Russian Federation (40.2/100,000 population) and Ukraine (37.1/100,000 population) in the Eastern European region, and Albania (3.6/ 100 000 population) and Turkey (3.4/100,000 population) in the Central European region. The total number of PrEP offering centers in the CEE region was 95. While 5/22 (22.7%) countries had no specific PrEP centers, the number of PrEP offering centers varied between 1 and 24 in the remaining countries; the highest numbers were reported by Ukraine (24), Poland (16), the Czech Republic (9), and Romania (9). Turkey and Albania with the highest rates of new diagnoses in Central Europe had no PrEP offering centers; Russia with the highest rate of new diagnoses in Eastern Europe reported only having 5 centers. The estimated access rate of PrEP showed great variation between countries, ranging from <1% (Albania) up to 80% (Bulgaria); 9 countries (40.9%) could not make an estimation for access rate. The number of people on PrEP in the region was estimated to be above 9000; 7 countries could not make any estimation and the highest numbers were reported from Poland (4000), Ukraine (2500), and Georgia (880). Overall, 16 countries (72.7%) reported that they were aware of informal/off-license PrEP use and 63.6% of respondents asked their newly diagnosed patients if they had ever used PrEP; the number of patients reporting prior PrEP use was very low, ranging from zero to five. Only two countries (the Russian Federation and Ukraine) reported being involved in PrEP clinical trials, both with long acting injectable drugs. The data for Ukraine were collected before the war. 

Barriers for access/wider access to PrEP use are listed in Table 2.

A comparison of responses from the countries that responded to both questionnaires in 2018 and 2021 (*n* = 20) revealed that PrEP licensure increased from 68.4% in 2018 to 75% in 2021, with 3 new countries (Lithuania, Turkey, and Ukraine) reporting PrEP licensure. Between 2018 and 2021, the number of countries prescribing PrEP free of charge increased from 10.5% (*n* = 2) to 25% (*n* = 5) and those prescribing PrEP within the public health system but the patient had to pay for the drugs increased from 21.1% (*n* = 4) to 30% (*n* = 6), respectively. Between 2018 and 2021, the number of countries where generic TDF + FTC was available increased from 20% (*n* = 4) to 35% (*n* = 7), whereas the number decreased from 25% (*n* = 2) to 15% (*n* = 3) for countries where both original and generic TDF+FTC was available, respectively. PrEP was not available for purchase at pharmacies in 8 (40%) countries both in 2018 and 2021. The median (IQR) price for generic TDF+FTC was 70 (33 to 200) and 50 (17 to 70) EUR in 2018 and 2021. The corresponding prices for original TDF+FTC were 292 (159 to 538) and 199 (95 to 249) EUR, respectively. There was a significant increase in the number of PrEP offering centers from 47 in 2017 to 95 in 2021, and the estimated number on PrEP increased from 4500 to 9000. The results of the comparison are summarized in Table 3. 

Non-occupational PEP was available in 19 (86.4%) out of 22 countries; it was reimbursed fully in 8 (36.4%), only for accidental exposures in 9 (40.9%), and was not reimbursed in 5 (22.72%). Non-occupational PEP was included in the guidelines in 12 (54.54%) countries. Occupational PEP was available in 21 out of 22 (95.5%) countries. Occupational PEP was financed by general insurance in 17 (77.3%), by additional insurance or the employer in 5 (22.7%), and was not covered in 1 (4.5%).

The summary of results for PrEP and PEP availability in Europe on a country basis is shown in Table 4.

## 4. Discussion

This study showed that although at a slow pace, PrEP roll-out was in progress in the CEE countries with significant achievements compared with the past, which is promising. Since its approval in 2012 by the US Food and Drug Administration, PrEP has become available on a global scale, offered as a registered strategy or in clinical studies or demonstration projects in 78 countries [16,27]. Europe has been hesitant to embrace and implement PrEP and France was the first country to rollout PrEP with full reimbursement in 2016, followed by a slow but steady scale-up mostly in the European Union or Western European countries, with a considerable delay in CEE countries [16]. In a similar previous survey from the region, PrEP was registered in only 34.2% of the countries in the CEE region in 2017, although healthcare providers reported that they were ready to recommend PrEP use to eligible persons if it was available [25]. A second study run in 2018 including the same countries showed that the number of countries offering PrEP in the region had doubled-up within a year [26], and this achievement was confirmed by recent data from the European Centers for Disease Control (ECDC) [17]. However, the COVID-19 pandemic is thought to have negatively influenced these achievements in the region [28].

PrEP uptake globally and regionally is still much lower than what is necessary to have a significant impact on limiting the epidemic. Reports from the European region call the attention to significant regional variability in the availability of and access to PrEP within a spectrum of low- to high-income countries and even in those with early and successful PrEP roll-out, including several EU countries [9,27]. Currently, Europe comprises only 7% of global oral PrEP initiations [27]. PrEP use per 100 000 adult population ranges between 0.5 to 3.6 in developed European countries such as Poland, Spain, Germany, France, UK, and the Netherlands, whereas it is 6.2 in yhe USA and from 6.2 up to 150.4 in several underdeveloped African countries [9]. A study from France, which was the earliest in Europe, in order to implement PrEP with significantly higher numbers on PrEP compared to other European countries, reported that people who are eligible, aware, and intending to take PrEP represented only 15.2% of non-users [29]. Another study looking at the PrEP gap in European and Central Asian countries reported that the overall PrEP gap was estimated to be 17.4%, ranging from 4.3% in Portugal to 44.8% in Russia [17], with a greater gap in non-EU or low–middle income countries compared with EU- or high-income countries, which underscores the striking diversity in the region. In addition, there was great variation between the number and rate of people using PrEP at least once, ranging from one (0.04/100,000 adult population) in Moldova to 9078 (52.5/100,000 adult population) in France [17]. Similarly, in our study, the estimated access rate to PrEP varied largely between countries ranging from <1% to 80% and the number of people on PrEP from zero to an estimated number of 4000. The most striking finding was that among the countries with the highest rate of new HIV diagnoses, Poland and Ukraine reported the highest number of PrEP offering centers, whereas Russia had only 5 and Albania and Turkey had no centers; in addition, Russia reported a very low estimated number of people on PrEP. 

Earlier reports from the region and our results suggest significant improvements in terms of PrEP availability and accessibility in CEE within the last five years with a steadily increasing number of countries offering registered PrEP, a higher number of PrEP offering centers, and larger number of people on PrEP compared with the past [25,26]. However, the numbers are still low and several barriers were defined that could impede wider usage of PrEP. In earlier reports, the major barriers for PrEP use were mostly system-driven, such as PrEP not being licensed, not being covered by the public health system, and lack of guidelines [25,26]. Previous studies showed that TDF and FTC have been available for antiretroviral treatment (ART) free of charge in all countries in CEE for many years, even before PrEP roll-out in France [30]. So, the main issue is unavailability of TDF and FTC for PrEP rather than ART. In countries where PrEP is still unlicensed, earlier fears for PrEP causing sexual disinhibition resulting in an increase in sexually transmitted infections, and resistance to antiretrovirals in case of an unrecognized acute infection, in addition to financial restrictions and lack of technical expertise to roll-out PrEP may be predicted to be the major barriers. Our study revealed that an improvement in PrEP registration led to a shift towards subject-driven barriers such as lack of knowledge among people in need of PrEP and low perception of HIV risk, with lack of reimbursement still being considered one of the main barriers. In line with this, the cost of the drugs in several countries is too high to pay out of the pocket, even for the generic formulations and the added cost of the follow-up tests further challenges the decision of an individual to use PrEP. Considering the likelihood of long-term PrEP use in a sexually active young individual, the lifetime economic burden would be unbearable. In a larger scale study by ECDC including 53 countries from Europe, among the countries where PrEP was not available and reimbursed, the major barrier to PrEP use was reported to be cost, and the majority of those that reported high cost defined it as a high- or medium-importance barrier [17]. Another study from the Netherlands suggested that men who have sex with men (MSM) with a better financial situation had greater PrEP uptake and that a drop in the price increased the uptake [31]. 

A low level of PrEP awareness and low perceived risk of individuals who are in fact at high risk of acquiring HIV infection is another challenge for PrEP implementation. PrEP awareness was shown to be related to residency, age, education level, and HIV risk perception [32]. Although awareness levels among white and educated MSM are usually quite high and risk level definitions are in line with their actual risk [33,34], heterosexual men and women do not usually consider themselves at risk of acquiring HIV infection. A study from the USA [35] analyzed heterosexual participants in a survey questioning risk perception reported that 84% of those with a high risk of acquiring HIV according to predefined HIV risk characteristics perceived themselves as having no or low risk, and consistent condom use was <20% in this group, despite a high knowledge level about HIV transmission. A large-scale survey including the general population in Britain suggested that the majority of those who reported practicing unsafe sex did not have a high perceived risk of HIV infection [36]. There are also disparities in PrEP awareness and risk perception among other vulnerable groups such as ethnic minorities, women, and displaced people [36,37,38]. People with lower education levels and low risk perception are reported to have significantly lower interest in using PrEP [33,35]. 

PEP is an emergency protection tool in the case of unexpected exposure to HIV. The average estimated risk of HIV transmission in the healthcare setting is reported to be 0.3% after a percutaneous exposure to HIV-infected blood and 0.09% after mucous membrane exposure [39]. Although strict adoption of universal precautions has decreased high-risk exposure to HIV in the last two decades, the prevalence may be much higher than expected due to possible underreporting [40,41]. Two systematic reviews [42,43] reported a high worldwide prevalence of needlestick injuries. Exposure to blood or body fluids in the healthcare setting were 56.2% and 56.6% during career time and 32.4% and 39% in the previous year, respectively. The pooled prevalence was highest in developed and lowest in developing countries. The evidence for PEP was derived from a very early case-control study where zidovudine led to an 81% reduction in the risk of acquiring HIV and currently is the only approach to prevent HIV infection after possible exposure [44]. Thus, the free availability of PEP for occupational exposure is a prerequisite for the protection of the healthcare provider. The majority of countries reported occupational PEP availability and full reimbursement by various modalities. However, a country that does not provide occupational PEP availability in 2022 should introduce this service without further delay. 

Although at a lower rate, non-occupational PEP was also available in the majority of countries. However, in contrast with occupational PEP, it was fully reimbursed in very few countries, probably due to concerns and evidence that it might create a false perception of security, leading to more frequent condomless sex and that PEP requests would rapidly grow leading to prescriptions for low-risk activities [45,46]. The use of PEP for the prevention of HIV infection in sexual exposure or among people who inject drugs has been a controversial issue over several decades. Although recent guidelines recommend the use of PEP in high-risk non-occupational exposure [47,48,49], it has not gained wide access and the level of awareness for PEP has remained low, even in countries where it has been available and fully covered for many decades [50,51,52]. Major barriers for low usage of PEP were found to be lack of awareness of healthcare workers, lack of awareness of those in need, lack of reimbursement, fear of side effects, and stigmatizing attitudes [53,54].

Despite several limitations such as the descriptive nature of the study and data collection depending on personal statement of the country representatives, this study is a snapshot of the situation for PrEP in Central and Eastern Europe, which is currently the only region with a rising HIV epidemic. Although much has been accomplished within the last few years in terms of PrEP availability and usage, there is great diversity between countries, regions, and settings, most likely due to variations in sociocultural characteristics, political structures, and available resources, which creates a major challenge for progress. 

## 5. The Way Forward

Evidence showing that PrEP is a highly effective tool in reducing new diagnoses, is critical to convince governments to roll-out/scale-up PrEP. Data from the region are very limited. Thus, the region needs collaboration between countries to generate data and analyze examples of best practice from neighboring countries. 

As there is no “one size fits for all” model for PrEP implementation, expertise sharing by Western or neighboring countries with previous experience on PrEP roll-out can help countries to develop their own models in line with their needs and resources.

Many countries in the region suffer from limited financial capacities. In this context, funding from international agencies seems to be the best solution to overcome barriers to the successful implementation of best practices.

Civil society organizations have a key role in conveying the needs of their communities to governments and to advocate for their rights to prevention from HIV without being subjected to stigma and discrimination.

## 6. Conclusions

Without optimistic expectations for an effective vaccine in the near future, PrEP is one of the most promising tools to help better control the epidemic. Therefore, faster scaling-up of PrEP in the region could accelerate achieving the UNAIDS goal of less than 200 000 annual HIV infections by 2030 and for this reason PrEP targets should be highly prioritized.

## Figures and Tables

**Figure 1 vaccines-11-00122-f001:**
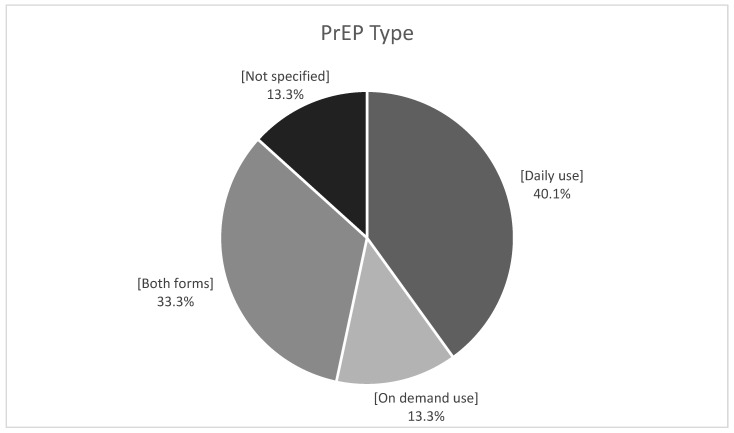
PrEP type available in countries (*n* = 15) where PrEP is licensed.

**Table 1 vaccines-11-00122-t001:** PrEP availability and accessibility in 22 countries in Central and Eastern Europe. Number of responses for all questions, *N* = 22.

PrEP Availability and Accessibility	*n* (%)
** *PrEP licensed* **	15 (68.2%)
** *PrEP in guidelines* **	
In national HIV guidelines	12 (54.5%)
Specific PrEP guidelines	1(4.5%)
No need for guidelines to prescribe PrEP	1 (4.5%)
** *PrEP prescription* **	
Free of charge within the public health system	5 (22.7%)
Free of charge within the public health system but patient has to pay for drugs	6 (27.3%)
Free of charge within the public health system but patient has to pay for drugs and tests	1 (4.5%)
In public practice	2 (9.1%)
Not prescribed	2 (9.1%)
Other settings *	6 (27.3%)
** *Formulations available in pharmacies* **	
Generic form	6 (27.2%)
Both forms	4 (18.2%)
Not available	10 (45.5%)
Available in other settings	2 (9.1%)

* Pilot programs, demonstration projects, clinical studies or purchased online.

**Table 2 vaccines-11-00122-t002:** Barriers to PrEP use.

Barriers	Percentage
Lack of knowledge/awareness among people who are in need	59.1%
Not being reimbursed	50.0%
Low perception of HIV risk	45.5%
Lack of knowledge among healthcare providers	40.9%
Fear of stigma and discrimination	31.8%
PrEP not being licensed	22.7%
Fear of side effects	13.6%
Lack of decentralization	13.6%
Other *	27.3%

* Shortage of healthcare providers, cost of diagnostics, referral system requirement, lack of medication, resistance of the Ministry of Health, legislative barriers, and high cost of drugs.

**Table 3 vaccines-11-00122-t003:** Comparison of responses given by countries that completed both surveys in 2018 and 2021. Number of responses, *n* = 20.

PrEP Availability, Accessibility and Use	2018	2021
** *PrEP licensed* **	13 (68.4%)	15 (75%)
** *PrEP recommended in national guidelines* **	10 (50%)	11 (55%)
** *PrEP prescription* **		
Free of charge within the public health system	2 (10.5%) *	5 (25%)
Free of charge within the public health system but patient has to pay for drugs	4 (21.1%) *	6 (30%)
Free of charge within the public health system but patient has to pay for drugs and tests	1 (5.3%) *	1 (5%)
In private practice	1 (5.3%) *	2 (10%)
PrEP is not prescribed	4 (21.1%) *	1 (5%)
** *PrEP availability in pharmacies* **		
Original form	1(5%)	0 (0%)
Generic form	4 (20%)	7 (35%)
Both original and generic forms	5 (25%)	3 (15%)
Not available	8 (40%)	8 (40%)
Other availability	2 (10%)	2 (10%)
** *PrEP offering centers* **	47	95
** *Estimated number on PrEP* **	4500	9000
** *Informal PrEP use* **	13 (68.4%) *	15 (75%)

* One missing answer.

**Table 4 vaccines-11-00122-t004:** Summary of PrEP and PEP availability on a country basis.

Countries	PrEP Available	PrEP Guidelines Available	PrEP Offering Centres Available	Informal PrEP Use	Non-Occupational PEP Available	Occupational PEP Available
Albania				√		√
Armenia	√	√	√		√	√
Belarus	√		√	√	√	√
Bosnia and Herzegovina		√	√	√	√	√
Bulgaria	√		√	√	√	√
Croatia	√		√	√	√	√
Czech Republic	√	√	√	√	√	√
Estonia	√	√	√		√	√
Georgia	√	√	√		√	√
Greece				√	√	√
Hungary	√	√	√	√	√	√
Lithuania	√		√		√	√
Macedonia		√	√			√
Montenegro				√	√	√
Poland	√	√	√	√	√	√
Romania			√	√	√	√
Russian Federation	√	√	√	√	√	√
Serbia				√		
Slovakia	√	√	√		√	√
Slovenia	√	√	√	√	√	√
Turkiye	√	√		√	√	√
Ukraine	√	√	√	√	√	√

PEP, postexposure prophylaxis; PrEP, pre-exposure prophylaxis.

## Data Availability

Data supporting the reported results can be obtained from the corresponding author upon request.

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
