# Peer review of "PrEP Scale-Up and PEP in Central and Eastern Europe: Changes in Time and the Challenges We Face with No Expected HIV Vaccine in the near Futureâ€"

_vaccines, 2023, doi:10.3390/vaccines11010122_

Round 1
Reviewer 1 Report
Authors present an updated review on impact of PrEP and PEP in central and Eastern Europe and ongoing challenges to access and use. Limits of study are due to only 1 report per European country for assessment of limitations and access. No good solutions are presented to increase access and availability to reach goal of 95/95/95. Granted there are a number of factors that can influence: politics, stigma, denial, lack of infrastructure within each country. Sadly if this is the state of central and Eastern Europe, can only imagine the challenges for access in African and other 3rd world countries.
Reviewer 2 Report
Dear Editor,
In this manuscript the authors describe how is the access to PrEP and PEP in different countries from Eastern Europe based on an online survey provided to relevant entities. The topic is interesting and the manuscript is overall well written. However, before acceptance, I recommend the correction of certain aspects:
- “Central and Eastern Europe is a long-neglected region in terms of addressing the HIV epidemic due to the long-term low prevalence of HIV infection, and economic, cultural, social and political diversity of the region.” Could you give examples, please?
- I suggest to use either mean +/- standard deviation or median (IQR), based on the non-/normal distribution of your data
- In Figure 1, the meaning of the different sections of the pie are missing
- “One country reported […]. Another country reported”. Please, provide the names of the countries in any case
- “The highest number of new HIV infections diagnosed in the previous year was highest”. Please, do not provide only absolute numbers, but also rates. According to your data, in Ukraine there are more infections than in Turkey, but the fact is that there are MANY more, as Ukraine has/has ca. 40 M inhabitants and Turkey ca. 70 M.
- Provide always n of the answers with the %
- No data is available on risk behaviours?
- Discussion: If FDA approved PreP in 2021 and France in 2016… I would suggest to rephrase, and provide your text in chronological order
- Why is PrEP not covered in some of the analysed countries? Why the PEP is covered only in work-related exposure? In countries where PrEP and/or PEP is not covered, is the regular HAART treatment covered? Could you hypothesize or give explanations to your results?
- Could you provide the survey as an appendix?
- Is it possible to provide a map with the most meaningful results for PrEP and PEP, respectively?
- How was determine who should answer the survey from each country?
